# Media as Religion. Stardom as Religion. Really?
## *Christian Theological Confrontation*

**Terézia Rončáková** 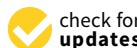

Department of Journalism, Catholic University in Ružomberok, 03401 Ružomberok, Slovakia;
terezia.roncakova@ku.sk

**Abstract:** In the more recent scholarly literature on media, pop culture or celebrity studies, there has been a growing tendency to identify media, stardom and other pop culture forms of cult with religion. An increasing number of concepts have sprung up such as "media as religion" or "stardom as religion". However, these concepts need to be critically scrutinized as to whether the use of specific theological terms in those concepts is sound and consistent—or, as the case may be, superficial. The primary aim of this paper is to examine whether there are essential intrinsic similarities between religion and media. To answer this question, we have examined the structural similarities between media and religion (by comparing their use of ritual and liturgy; emotions; cosmology; myth and archetype; and the cult of individualism in particular). Subsequently, we have analyzed the key terms that have emerged from those comparisons (religion and faith; God; emotions; community; liturgy; cosmology; archetypes; saints; individualism). The term religion is used in its broad sense; however, the subject is examined in detail within the context of Christian theology. We came to the conclusion that media religion is a non-theistic religio without God, with an exclusive emphasis on social cohesion. The absence of verticality, lack of transcendence to eternity as well as the non-existing relationship with God as a person—have determined the remaining partial conclusions presented herein.

**Keywords:** media; religion; stardom; ritual; liturgy; emotions; myth; archetype; cosmology; cult of individualism

---

## 1. Introduction

The focus of research on the relationship between media and religion has gradually shifted from the effort to describe how religion, or more specifically, the church use media to transmit religious messages—by exploring the key characteristics which affect the transmission of such messages through mass media—to a more profound attempt to explore the underlying relationship between religion and media. One way to approach the subject is to examine these phenomena from the perspective of the anthropology of religion. This approach focuses on the inherent systemic similarities between the nature of religion and media. G. Thomas (2005) discerns four levels of interaction between media and religion: (self) presentation; use of religious motifs and symbols; the ritualization of television broadcasts and cultural intervention. Cultural intervention is not just about individual motifs or television broadcasts, but the entire television system, including its individual as well as social functions similar to those typical of religion. N. Couldry (2003) also shifted his attention from individual media products and religious events to the actual existence of media and the intricate ways in which they transform the society due to their ritualistic nature, while P. Lardellier (2005) uses the term ritual media. Similar speculative inquiries can be found in the field of celebrity studies where the characteristics of religion are ascribed to stardom (Boyd 1958; Laderman 2009), hence the term "stardom as religion" (Frow 1998; Rojek 2001, 2012). These tendencies within the context of celebrity culture have been examined in detail by P. Ward (2011, 2017, 2019) who suggests a conciliatory concept of "pararaligion".

This leads to the key question posed by J. Derrida (2001, p. 69) as to whether there are intrinsic similarities between religion and media technology; or, whether there exists something like the "religion of the media". In that respect, we will examine the difference between the understanding of religion, faith, God, rites, liturgy, unity, community and sainthood within theology and compare them with the concepts found in theories of media anthropology. This also leads to the question of the role of emotions, spiritual search, individualism or archetypes within the framework of theological thought and within the concepts of media anthropology. By clarifying these terms as they are understood within the two realms, we can then assess the extent to which the interpretations of these terms converge or diverge from each other, and evaluate how (theologically) reasonable it is to ascribe the essential characteristics of religion to media (or other phenomena).

As part of our endeavor, the term "religion" will be used in its broader sense which also encompasses other religious traditions; however, the subject will be examined in detail within the context of the Christian religion and Christian theology. When we speak about a phenomenon (whether it is the media or celebrity culture) as something akin to religion or having the characteristics of religion, it is reasonable to explore the terms and concepts used in such an experiment through the lens of one of the biggest traditional formal religions. This is not undertaken in the sense of a confrontation or an attack but in the form of a curious inquiry with a focus on the question of the extent to which the terms adopted from religion and used in the new contexts converge toward or diverge from their original meanings—regardless of whether they are used in a figurative sense or in their full original sense. It should be noted that the recent concepts of media or celebrity culture "as a religion" have originated in the West; the terminology they use is then quite naturally inspired by Western Christianity, and therefore it is reasonable to make our case within the context of Christian theology. In that respect it may be worth noting that J. Derrida developed the theory of a fundamental bond between media and Christianity, which—unlike Judaism, Islam or Buddhism or other religions—was literally born to become subject to publicity and mediation; in that sense, Derrida argues, Christ was the first journalist (Derrida 2001, p. 58).

Moreover, our deliberate confessional focus employed in this exercise needs to be set into the mainstream of theological thought and cannot engage in the intricacies or nuances of the internal discourse within the field of theology, otherwise, the outcome of our exercise would become rather vague. Therefore, this study will refer to evangelical conservative thinkers only to some extent and draw primarily on Catholic theological thought including the official document of the International Theological Commission established by the Vatican. This also means that for the purpose of this study, terms such as theology, Christianity and religion should be conceived (unless the context implies otherwise) within the framework of the mainstream conservative Christian or—more specifically—the Catholic creed. Hence, we are exploring the subject from the Catholic theological perspective since that is the theological underpinning of the primary target audience for our theoretical analyses, arguments and findings.

## 2. Structural Similarities between Media and Religion

S. Hoover and K. Lundby showed in (Hoover and Lundby 1997) that the media used religion more frequently than religion used the media. Their pioneering work *Rethinking Media, Religion and Culture* sparked scholarly interest in the subject of the relationship between media and religion and points to the fact that aspects of media consumption and representation appear to carry religious significance in themselves, a notion which was also referred to by P. Ward (2019, p. 13). S. Hoover (2001) suggests that the natures of both religion and media are undergoing change and that they are, in fact, converging in important ways. S. Hjarvard examines in detail the impact of media on religion and concludes that the media are linked with these institutions to such an extent that in a sense they impose new rules on their functioning and "alter the ways in which people interact with each other when dealing with religious issues" (Hjarvard 2008, p. 11). This process of mutual convergence between religion and media seems to be bidirectional, with the media taking on religious

characteristics and ritualizing their discourse and modes of collective participation (Lardellier 2005, p. 70). M. Coman (2014) pointed to two plausible comparisons: religion as media (mediating the contact between mankind and transcendence) and media as religion (this notion is based primarily on the presence of archetypes in media). In his elaborate linguistic analysis of religious communication, J. Mlacek (2012) concluded that the profane media not only borrows some partial metaphors from the sphere of religion, but it also takes on the entire structures and texts clearly associated with religion, as well as various elements from other semiotic non-verbal means of communication. These attitudes were succinctly summarized by G. Thomas:

> "Television is not just a sign system people use to interpret their lives. It is not just a cultural system of symbols per se. Television in late modern societies is not just a text rewritten by every viewer. ( … ) within the stream of television, forms of communication emerge that bear a striking resemblance to well-known religious forms". (Thomas 2005, p. 88)

Many of these theoretical concepts of media and religion were developed at the time of the dominance of television and based on research into television institutions, products and audiences. Today, we are witnessing the emergence of the era of new media (Campbell 2010, 2013; Campbell and Rule 2020), but it is nonetheless perfectly legitimate to draw on, re-evaluate and challenge the validity of the theoretical concepts formulated in the past. Therefore, this study generalizes the two subordinate categories: television and media. Therefore, for the purposes hereof, television is conceived synecdochically as "pars pro toto". This is in line with the approach used by the authors of the original television-based research studies. Here, it is enriched with the implications of the new media reality which opens the door to new research efforts on the subject.

P. Ward (2011, pp. 2–4) suggests that the religious meaning of celebrity culture in the academic community is regarded as common and that it is thought to be a consequence of the spiritual collapse of our modern society. The spiritual vacuum has been filled in by media, primarily by specific figures—the celebrities. "The connection between celebrity and religion is more than a discursive trope in popular media or in fan cultures. In academic literature it has become commonplace interpretative category" (Ward 2017, p. 3).

The structural similarities of media and religion have been elaborated in detail within the current theoretical concepts in four areas:

- ritual and liturgy,
- emotions,
- cosmology,
- myth and archetype.

*2.1. Ritual and Liturgy*

The comparison of the media system and program to liturgy is based on the understanding of media from the perspective of anthropological considerations on religion. G. Thomas (2005, p. 79) elaborates on this perspective using the following metaphor: media theorists take on the glasses (i.e., the theoretical apparatus) from anthropological considerations on religion and then go ahead and watch the television shows.[1] Through this approach such theorists can identify unambiguous elements of rites and then term these "media rites" in the figurative sense as "profane liturgies" (Rivière 1995).

The notion of the ritual mode of communication was pioneered by J. Carey (1989) and later elaborated by other scholars (including Andok 2017; Sumiala 2012 or Couldry 2005). At the foundation

---

[1] Similarly, as part of our investigation, we have taken on the "glasses" of Christian theology in order to take a closer look at media theories. To the objection of mixing two incompatible discourses, we share the opinion of G. Thomas that "fruitful research programs sometimes benefit from a risk-taking strategy. They derive their advantage from an imaginative and challenging yet methodologically controlled crossing of borders between traditional academic disciplines" (Thomas 2005, p. 79).

of N. Couldry's concept is "the myth of mediated center", that is, the "the belief or assumption, that there is a center to the social world and that, in some sense, the media speak 'for' that center". Whereas the religious ritual refers to something transcendent, the media only refer to themselves as the interpreters of the "social center". According to Couldry, media rituals cannot be reduced to Neo-Durkheimian large-scale television media events (Dayan and Katz 1992). For Couldry, media rituals are situations in which people look up to the media as their natural connection with the social "center".

> "Studying media rituals in this nonfunctionalist way is the opposite of isolating particular moments and elevating them to special, even 'magical', significance. On the contrary, ( … ) the transcendental 'value' associated with 'the media' is based on their presumed ability to represent the social whole. ( … ) Media rituals (in the sense in which I am using the term) are actions that are capable of standing in for wider values and frameworks of understanding connected with the media". (Couldry 2005, p. 64)

The media themselves represent the social itself; they are our "natural" access point to social space. This can be seen in the kind of "pious" behavior which people outside the media exhibit when they find themselves in the media, for example when they are invited to a talk-show or to a general knowledge quiz or other television contests in which they meet media celebrities either accidentally or as part of large-scale events. A further example of this is when people feel a kind of piety when visiting places somehow associated with the media.

G. Thomas (2005, p. 82) concludes that ritual forms that are deeply grounded in anthropology and are, to a certain extent, universal become structures for media formats; television thereby absorbs and modifies forms from the religious field. He considers liturgy to be one of the basic religious aspects of television (media) where it takes on the form of rhythms, cycles and repetitions. "With its reliability, its stable forms, well-known figures and familiar shapes, the endless liturgy satisfies a need for continuity and intimacy" (ibid., p. 86). Independent of the viewer's thinking and feeling, it differentiates television from the fragility of communication based on interaction. The inability of viewers to disturb or influence the stream resembles the fixed nature of the communication process in traditional religious rituals. It connects the viewer with some kind of "world beyond" their everyday life; it facilitates an entry into so-called prefigured realms of experience and offers "small-scale transcendences" that are "always at hand".

*2.2. Emotions*

Emotions are inherently associated with the concept of the ritual and "liturgical" nature of media because according to a number of theorists, it is the emotions that integrate the heterogeneous and fragmented mass audience without any alternative of direct interaction into a kind of "diasporic community" (Dayan and Katz 1996). According to P. Lardellier, the cornerstone of these communities is the "common gaze"—they link "seeing" and "seeing with", and allow collective participation (Lardellier 2005, p. 70). To describe a participant of the community ritual, P. Lardellier borrows a term from cinema theory; his "spect-actor" is someone who both views and at the same time plays a role. Theories related to media emotions are based on the pioneering work of Dayan and Katz (1996) and the concept of "media events" which they developed. They suggest that a television event multiplies by 10 the effect of traditional ceremonies. According to P. Lardellier, it was in the televisual sphere that the metaphor "urbi et orbi" actually became reality, and where the term "orbi" has indeed taken on the meaning of "to the world" (Lardellier 2005, p. 75). Analogically, thanks to the "miracle and vertigiousness of the ritual media", a large part of the planet was able to attend the funeral of J. F. Kennedy on 25 November 1963 as people "communed together on the altar of emotions, tears and history" (ibid., p. 77):

> "Diasporic community is evanescent and virtual, indeed, but oh, how powerful and dense!
> For it is masterfully welded and bound by an emotion, a memory and a common gaze".

Commonly experienced emotional ecstasy, in a sense, enables individual members of the audience to "touch" the mediated event. The transmitted piece of information takes on a "haptic and emotional nature" (Rusnák 2013, p. 43). The media standardize our reality and make the world more predictable; they stereotype emotions (which, in reality, are much more diverse). They transform actual reality into a kind of media reality, which is not "experienced" but "shown" (ibid., p. 39). While M. McLuhan discerned two key historic stages of media development—the "ear-dominant period" and the "eye-dominant period"—J. Rusnák suggests that these will be followed by the "haptic period", in that the collectively experienced emotionality will constitute a new feeling of a "virtual tribe". In this stage, a new kind of media audience (here and now) will be born.

## 2.3. Cosmology

From the perspective of religious studies, transformation (from uncertainty to certainty), interpretation and integration (Zaviš 2008) represent the key functions of religion. In today's fragmented world full of various social subsystems and lifestyles, where "normative contexts are relativized by recognized plurality" (Thomas 2005, p. 86), sectors of reality beyond direct everyday experience appear increasingly opaque for more and more people. The media system produces a comprehensive self-description of modern society which has at least partially replaced traditional religious cosmologies in many respects. Religious cosmologies serve, firstly, to integrate a plurality of world aspects into a single interpretative horizon via a unified and comprehensive description given by a "privileged observer". This epistemological "God's-eye view" (ibid.) is currently delivered by the media—and creates a situation where the media take on the function of a "sacred canopy" (Berger 1967).

However, such a media cosmology fails to appreciate the essence of things and changelessness. On the contrary, it only includes what is interesting and at the moment. Not the substance, but only the temporality (actuality); not the unchanging, but that which creates resonance (the interesting) are the building blocks of this cosmology (Thomas 2005, p. 87). J. Rusnák points out that for some, media mimesis plays even a more important role in their lives than realitas (Rusnák 2013, p. 41).

## 2.4. Myth and Archetype

Some theorists place the myth at the "center of social life". J. Lule (2005) suggests that in today's world, it is news coverage that is closest to the "social center". Myths are stories of public interest, archetypal stories that play crucial social roles. They explain origins, promote order and represent social beliefs and values. Moreover, they carry a moral message—J. Lule calls them "moral tales"—because they provide examples of good, evil, right, wrong, bravery and cowardice. They help people to engage, appreciate and understand the complex joys and sorrows of human life (ibid., pp. 102–3).

J. Lule (2001) suggests there are seven mythical archetypes reflected in news coverage:

- the victim,
- the scapegoat,
- the hero,
- the good mother,
- the trickster,
- the other world,
- the flood.

M. Coman (2005, 2014) develops this idea further and examines whether mass media—as a "system"—also take on the functions of institutions which have previously generated and spread myths and mythologies under the "old model" of societies; or whether the mythical dimension is present only in certain specific situations as a typical feature of individual media "products". In other words, whether mythical properties constitute the "essence" of media discourse or they only represent its "accidents". This distinction points to two alternative models:

1. "the bazaar model": journalists pick and choose from the symbols on offer more or less randomly and frame the current events on an ad hoc basis;
2. "the shrine model": media deliberately ritualize their output and confirm their own authority.

Under the bazaar model, news people operate as "cultural bricoleurs" (Newcomb and Hirsh 1985) and they can be compared to the "primitive empiricists" mentioned by Claude Levi-Strauss: they test reality by experiencing life, not by structuring the experience of life in a formal, systematic way. For the shrine model, meanings are presented as given (top-down) and their mediators as messengers of a transcendent truth (Coman 2014). Thus, the media "mobilizes the sacred" and becomes the "guarantee of ontological security" (Coman 2005, p. 117). Within the context of celebrity culture, one can find references to mythology in the work of, for example, G. Landerman, who sees celebrities as semireligious figures or false gods, arguing that religious practices associated with celebrity worship invest in "mythologies that promise immortality" (Laderman 2009, pp. 64–65).

## 3. Cult of Individualism

What is the object of worship for the media? Whom do they serve? What kind of god is the god of media religion? E. W. Rothenbuhler provides some plausible answers to these questions in his "Church of the Cult of the Individual" (Rothenbuhler 2005, p. 98). For Rothenbuhler, this role is played by the individual. He then refers to E. Durkheim ([1898] 1994) and his concept of the cult of the individual as the religion of modern society.

> "Durkheim's great contribution was to identify the structures and processes underlying modern, secular life that operate by the logic of religion but without its name". (Rothenbuhler 2005, p. 98)

If this is true, says E. W. Rothenbuhler, and if the cult of the individual is indeed the religion of modern society, then media is the temple of this cult.

> "If, as (e.g., Goffman 1959, 1967) showed, the person, the self, which is the sacred object of the cult of the individual, is a symbolic entity that can be known only through communicative processes, then the ceremonial practices of that cult (religion) must be in communicative processes". (Rothenbuhler 2005, p. 91)

The media system of consumer culture and celebrity has grown, according to Rothenbuhler, into a church of the cult of the individual. In consequence, "one of the media's most important activities is the production of saints and heroes, devils and ghosts, choirs, preachers, mullahs, gurus and bishops" (ibid., pp. 92, 94). As myth and ritual do more than present heroes and villains; it is the means by which heroes and villains are created.

> "Politics becomes (is) marketing. All of the institutional spheres differentiated in the evolution of modernity become more like communication industries. No one is in favor of it, and yet all participate. How else to explain it except as due to the inexorable, arational pull of religious faith? We must be our self and our self must be served. Attention to the self spreads, contagious and attractive as a religious ecstasy". (Rothenbuhler 2005, p. 99)

The cult of the individual has also been discussed by other scholars (Giddens 1991; Roof 1999). S. M. Hoover and J. K. Park have expanded upon Helland (2000) concept of "online religion" (which—unlike "religion online"—is not only a place of religious presentation but also at the locus of religious practice) and examined this "marketplace of symbols and contexts, where there is no priesthood, no dominant tradition or doctrine. There are no barriers to entry or participation" (Hoover and Park 2005, p. 250). Both conclude that online religion requires a process of seeking. However, their research also found that "seekers" rooted in one of the churches with a solid religious identity do not tend to look for their spiritual footing on the internet. Online religion is most suited for so-called "new age seeking",

with seekers creating hybrid traditions that combine Christianity with Eastern or Native American traditions. These seekers create their own "cocktail using ingredients from a variety of spiritual traditions" (Juhásová 2010).

With respect to individualization, P. Ward evens speaks of a "shift in the nature of the sacred in contemporary society", or a fundamental "subjective turn in contemporary religion" (Ward 2017, p. 8). This is a shift toward the self, as the primary concern and the central project of life, and it has become the new sacred; indeed, religion in the West is perceived as a resource for the project of our self. One of the primary resources used in creating the project of one's self is celebrities—specifically, celebrities whom we judge aesthetically and morally or compare ourselves with; the celebrities who become our inspiration or those whom we reject (Ward 2019, p. 4).

## 4. Analysis of Key Terms

When the structural similarities between religion and media described above—as identified through the lens of media studies and anthropological considerations on religion—are also examined from a theological perspective, several key terms need to be clarified upfront as part of an exercise set up as a confrontation between media and theology. Therefore, the following passage will discuss terms such as religion, faith, God, communion, unity, emotion, liturgy, archetype, cosmology, social order, sainthood and individualism.

### 4.1. Religion and Faith

From the standpoint of religious studies, it is quite difficult to define the term religion as such. "Existing definitions usually emphasize the relation between human subject and deity, a relation between the sacred and the profane, the immanent and the transcendent" (Nguyen et al. 2020, p. 139). Q. H. Nguyen et al. discern between secular religions (such as Confucianism) and more mystical forms of religion (such as Taoism or Buddhism). J. Ratzinger makes a further distinction between theistic and non-theistic forms of religion (Ratzinger [1968] 2007, p. 12). Sociologists and political scientists provide a variety of constricted definitions of religion: Hegel and Marx, for instance, considered religion as merely a system of laws, canons and thoughts; Weber analyzed the term only through its external forms, that is, human behavior or the lifestyles of believers; Tillich and Huntington saw religion as the essence of culture, and culture itself as a form of religion (Nguyen et al. 2020). According to P. Ward, it is problematic to attempt any kind of common definition of religion, but most approaches to religion involve at least one of the following ideas:

- a belief in a supernatural power,
- the significance of religion in generating community life or some kind of church,
- the influence of a divine power on people's lives.

On all of the points outlined above, celebrity culture fails significantly or does not meet what is required from a formal religion (Ward 2017, p. 8). Christian theologians jointly identify religion with faith—religion and faith (or belief) are typically regarded as synonymous (Nguyen et al. 2020; Krapka 2000; Ratzinger [1968] 2007).[2]

Faith is man's response to God; an acceptance of light, an expression of adherence, subjection and dedication to God. Religion is a journey to the light—to God (Krapka 2000). "Credo" contains a clear decision with respect to reality as such, it has a profound impact on how man sees the world and "it signifies, not the observation of this or that fact, but a fundamental mode of behavior toward being, toward existence, toward one's own sector of reality and toward reality as a whole" (Ratzinger [1968]

---

[2]　The Christian idenitification of religion with faith is in congruance with the main stream of conservative theological thought; a detailed analysis of alternative theological views is outside the scope of our investigation.

2007, p. 36). H. Küng presents faith as trust; whereas the world puts pressure on man to perform and derives his value based on his performance, God liberates him from this pressure and "vindicates him".

> "If a human being wants to attain self-realization, freedom, identity, meaning and happiness as a person, he can only do so in unconditional trust to the one, who has the capacity to give all these things". (Küng [1974] 2007, p. 443)

One of the essential characteristics of faith is the immediate closeness to an imperceptible being—"something completely different from what is commonly regarded as monotheism" (Buber [1986] 2007, p. 195). God is "Thou", it is "my God to whom I relate with the immediate closeness of "emuna", the love with all my heart and with all my soul and with all my might" (ibid.). A. Draguła (2013) differentiates between a "discourse of faith" which is unthinkable without the personal encounter, internal transformation and passion and a "religious" or "meta-religious discourse" which is typically associated with indifference and an absence of engagement.

So in the light of the above, is it possible for a man to relate to the loving God, to dedicate one's self, trust in Him with love and enter into an intimate relationship with God—within the realm of media religion? Clearly, none of this can reasonably occur within the framework of media religion; this kind of faith is simply not present. This supports the notion that media "as religion" is something akin to religious forms without faith; something much closer to the Roman concept of religio as a religious feeling understood mainly as the observance of certain ritual forms, in which the act of faith might be completely absent (Ratzinger [1968] 2007, pp. 34–35). T. Halík (2004) adopts a functional perspective in which it is not religion that integrates society, but that which integrates society becomes its religion. The word religion can be replaced with the Latin equivalent religio to express the power of building social cohesion. "After the Second World War, the media are growing and adopting the role of a force that integrates society" (Guzek 2019, p. 93).

*4.2. God*

The question of faith is closely related to the question of God. If one looks at faith (religion) as man's answer to God's calling and a relationship of love of God, then the term faith is unthinkable without its object—that is, God. Christian theology emphasizes God's initiative toward man, "eudokia": God's interest in man, that is, the way in which God looks at man. God's name is I AM and the human response is HERE I AM. A man stands before God and enters into the I–Thou relationship; in this situation, it would be disrespectful to ask "what is God like?", but the question that should be asked is rather "who are you, my God?" (Krapka 2000, p. 35).

J. Ratzinger raised an important point about modern religion without God. After the fall of the communist regime in Europe, "in the leaden loneliness of a God-forsaken world, in its interior boredom" (Ratzinger [1968] 2007, p. 11), it seems that religion became modern again and emerged from the ghetto of the subjective and private sphere in which it had been pushed by the 19th century—with the caveat that God had been forgotten. God has not been negated but is left instead as God with nothing to do. When it comes to the task of transforming this world, God is considered impractical. And "when God is left out of the picture, everything apparently goes on as before . . . " (ibid., p. 10).

But how is this related to the concept of media as religion? Do media guide us to God? Do they lead us to His presence? The answer, clearly, is no. Media communication is not vertical but perpetual: the media point to themselves as the interpreters of the "social center" and media rituals are situations where people look up to the media as the interpreters of the "center" (Couldry 2005). An external authority claiming the highest and definite truth is foreign to media religion. (Hoover and Park 2005). People who can easily accommodate online religion are typically averse to authority; they want to choose their own symbols, values, associations and ideas in order to construct their own identity and their own spirituality. "An institution is inconvenient, and dogma bothersome" (Ratzinger [1968] 2007, p. 11). Celebrity worship does not relate to the transcendent divinity, but to the sacralized "self" (Ward 2017). Viewers do not come to stand before the face of God in order to listen and submit themselves

in the "obedience of faith" (TT³, art. 11), but they are in search of their own face. They do not trust ("believe in") the one, whose appreciation, at the end of the day, is the only appreciation that counts (Küng [1974] 2007).

The media serve the social order, they serve themselves and they serve man as an individual—but all of this without God, without any notion of the Authority and, first and foremost, without the Truth.

*4.3. Emotions*

Obviously, emotions are perhaps the most visible feature of religious experience. The theoretical concepts of media as religion—or, for that matter, as some other phenomena resembling religious experiences—quite naturally embrace emotions, specifically the emotional ecstasy presented "as religious". From a theological perspective, emotions remain a marginal topic with no significant impact on the "response to God's calling" or the "decision" to believe. On the contrary, theologians emphasize that spiritual experience embraces man in his totality: his mind, his will, his memory and his emotions (Gavenda 2016). However, as pointed out by J. Ratzinger, within the current trend of "rediscovery" of the mystical aspect of religion, people strive for experiences and direct contact with the "entirely Other". Ratzinger agrees that, for example, in large-scale events such as World Youth Day, faith becomes an experience and something like ecstasy in a positive sense, although it is often a momentary experience. (Ratzinger [1968] 2007, p. 12). I. Bradley speaks of the new touchy-feely up-front spirituality of post-modern society with its search for values, its quest for experience and sensation and its openness to a whole range of visual and aural stimuli (Bradley 2004, p. 19).

Emotions in media turn religion into banality (Hjarvard 2012). "The concept of banal religion clearly ignores the key perspective of metaphysics for religion and the order of life. The supernatural space is replaced by action, emotions and acted contexts" (Guzek 2019, p. 89). However, one can also hear voices with a positive theological approach to emotions. M. Gavenda incorporated emotions as the key ingredient of his theory of "spirituality of the heart" which he suggests as a cure for the degenerative effects of media on humanity. Gavenda regards feelings as a form of "vehicles for transcending man from the realm of the sensory directly into the realm of the spirit" (Gavenda 2016, p. 52). They have the potential to draw man to the depth and essence of his existence and enable him to contemplate, reconnect and reenter into a profoundly existential dialogue.

*4.4. Community*

All of the community-related concepts—such as the (religious) capacity of media to unite people within a shared experience and to create kinds of diasporic communities (Dayan and Katz 1992), "virtual tribes here and now" (Rusnák 2013, p. 46) or ritual communities amalgamated by common gaze (Lardellier 2005)—play an important role within the theoretical frameworks of the various concepts of the religious nature of media. These concepts also point to a parallel with the Church as a community of believers. To a great extent, theology also embraces the aspect of community and unity. "Faith is at the same time a reality profoundly personal and ecclesial. In professing their faith, Christians say both 'I believe' and 'We believe'" (TT, art. 13). The individual is regarded as a fundamentally relational being; man is called into the community amalgamated by love to imitate the *agape* as a communion between the persons of the Holy Trinity (Valco and Sturak 2018).

Within Christian theology, the default understanding of the community of believers is essentially set within the context of the communion with the Triune God.

---

3    An abbreviation which stands for the document of the International Theological Commission, part of the Roman Curia: *Theology Today: Perspectives, Principles and Criteria* (International Theological Commision 2012); in references to the document, the article is provided instead of the page and the abbreviation TT is used.

"The Mystery of God revealed in Jesus Christ by the power of the Holy Spirit is a mystery of ekstasis, love, communion and mutual indwelling among the three divine persons". (TT, art. 98)

Thus, believers strive to attain unity because they are loved by the same Father and together, they form the one mystical body of Christ, that is, the church. A man who has the privilege to participate in the triune communion of love, is a man in the image of God (Imago Dei); a partner in dialogue with God, thereby enjoying all the dignity and rights which are a "reflection of God's glory; they are not achievements of man, but they represent the glory and honor bestowed upon man by God" (Krapka 2000, p. 58).

Based on the above one can clearly see the contrast between the ephemeral diasporic ad hoc communities created by media and the communion with God transcending to eternity within theological discourse. There is no regularly gathering community of celebrity worshippers; they do not constitute a religious community. (Ward 2017, p. 8; 2019, p. 3). Media do not seek love or eternity; they strive to arouse ephemeral emotions that are bound to a specific media event.

### 4.5. Liturgy

The concept of community is closely related to that of liturgy. The theological understanding of liturgy is rooted in the gathering of believers united in the love of Christ (Krapka 2000, p. 93). Liturgy is the common prayer of the Church with believers assembled to serve and worship the Father and experience deeply the love of the Holy Spirit (ibid., p. 85). M. Gavenda understands liturgy as "ontologically the most intense form of encounter" (Gavenda 2016, p. 174), because when the Church—the mystical Body of Christ—gather to pray, it is, in fact, Jesus Christ who is praying. Hence, liturgy is not just a cult or celebration of God, but something that extends the presence, effect and the engagement of God-man, Jesus Christ, who is actually and substantially present in the sacrament of the Church. Through the prayer of the Church, particularly when set into the liturgy, believers experience, learn and immerse themselves into the mystery of the Fatherhood of God. Together with Jesus, believers can utter the word Abba . . . "The essence of the mystery of our redemption lies in the fact that we are allowed to—and that we dare to say . . . " (Krapka 2000, p. 66).

What is the understanding of liturgy within the realm of media religion? Media liturgy incorporates liturgy as a rhythm, system, repetition, unchangeability and certainty (Rothenbuhler 2005). No aspect of the "ontologically intense encounter" is present. Within the Christian liturgy, the rite is considered one of the symbolic means of expression. In media, the rite is the goal, or more precisely, the object of "reverence" or adoration of the media—as the interpreters of the "social center". Within Christianity, on the other hand, the One who is the source of all the gifts is the object of adoration; and the following of Christ—perceived as a gift filled with joy and a sense of mission—naturally evolves into the adoration as such (Krapka 2000, p. 78). However, since neither God nor faith—as a response to God's calling—are present in media religion, media liturgy is void of this dimension and results in formalism.

### 4.6. Cosmology

Cosmology as a discipline of the understanding of the universe in its totality refers to the theology of Creation and the Truth in a religious sense. Within Christian theology, "the ultimate mystery of the nature, the world, the universe and man—will never be revealed in full by science (science can only confront man with that mystery), but only through Jesus Christ: it is the mystery of the love of God" (Krapka 2000, p. 53). "Everything in this world is created for man so that he always seeks the kingdom of God and God's justice in everything—in all the created things, all human beings, all the situations and events" (ibid., s. 55).

The relativization of the truth is foreign to Christian theology; theology refuses the secular contempt for anything that resembles the "ownership of truth". "Convictions come from truth and lead to it" (Ratzinger [1968] 2007, p. 12); to believe means to "participate in God's truth" (Krapka 2000, p. 72); " . . . the divine revelation the Church accepts by faith as universal saving truth" (TT, art. 5).

So how is the Truth reflected in the cosmology of media religion? Within the realm of media theory, the relativization of normative contexts and the preference for plurality is accepted as inevitable. The cosmological effect of media leads to the creation of a "privileged observer", a figure who mediates a unified and comprehensive self-understanding to modern societies (Thomas 2005). In that respect, metaphors such as "God's-eye view" or "sacred canopy" are suggested. But again, all of these considerations point to the "myth of a mediated center", and although the "center" is rather vaguely defined, what is undoubtedly meant here is definitely neither God nor Truth with a capital T.

### 4.7. Archetypes

One of the key aspects of the intrinsic similarity between media and religion is the presence of archetypes in media products and the capacity of the media to determine what is good or bad, right or wrong, beautiful or ugly. Religion is the natural home and the source of archetypes with a biblical background (e.g., sacrifice, the sacrificial lamb, the plagues of Egypt, the golden calf, the forbidden fruit, the lion's den, the massacre of the innocents). However, theology does not focus specifically on archetypes; they are regarded as being secondary and reflected upon within the context of social sciences.

Theorists discussing archetypes as religious manifestations of media often refer to C. G. Jung (1964, 1976) and his "collective unconscious" (Lule 2005). Jung himself concludes in his work Christus als Archetyp (Jung [1983] 2007) that a specific rabbi named Jesus was assimilated through the common archetype of a denied and perfect hero because "das Selbst" (the self) in the soul of each man looks for a symbol of a "more comprehensive wholeness than his own self" (ibid., p. 218).

Other theorists see archetypes as story values, a bazaar of values open to random selection or a shrine in which values are selected and deliberately offered to confirm the authority of media (Coman 2014). Admittedly, the media do not create their own archetypes but only borrow them from religion. The capacity of the media to determine good and bad does not originate from themselves—that is a power which is limited only to God through the ability to calm the storm, heal the sick, expel evil spirits, raise from the dead; God as the one who has "the authority, and not as the scribes" (Mark 1: 22). In that sense, the presence of archetypes in media does not prove the religious nature of media, but rather the power of religion and a spillover of religion into media.

### 4.8. Saints

Another suggested similarity between media and religion is the parallel between celebrities and saints. According to E. W. Rothenbuhler, celebrities are "saints" of the cult of the individual and one of the most significant activities of media is their production. Theorists are far from ignoring the degeneration of these "saints"—from heroes who find fame for their accomplishments to celebrities whose fame is itself their sole accomplishment (Boorstin 1961). Celebrities present themselves as if they are heroes regardless of the reality; they are products of the star system and the star-making machinery; products of the rationalization of the industrial production of culture (Rothenbuhler 2005). Nevertheless, scholars still point to the parallels between the qualities ascribed to specific celebrities and religious figures and also those between the spiritual experiences of members of celebrity fan clubs and the community of believers (Turner 2014). C. Rojek talks about "the commodified magnetism that celebrities possess with a performance culture that routinely trades in motifs of unity, ecstasy and transcendence" (Rojek 2012, p. 121).

Within the context of Christian theology, a saint is one who "simply takes the first commandment seriously: you shall love the Lord your God with all your heart and with all your soul and with all your might (Deuteronomy 6: 5)" (Guardini [1981] 2007, p. 382). Since God's revelation attained perfection in Jesus Christ, saints reveal the face of Christ by adhering to the beatitudes articulated by Christ: blessed are the poor in spirit, blessed are those who mourn, blessed are the meek, blessed are those who hunger and thirst for righteousness, blessed are the merciful, blessed are the pure in heart, blessed are the peacemakers, blessed are those who are persecuted because of righteousness (Matthew 5: 3–10)

(Krapka 2000, p. 105). R. Guardini suggests what an attentive observer should notice when meeting a saint: "quiet freedom, peaceful confidence in the search for meaning and direction, calming joy despite all the worries and troubles" (Guardini [1981] 2007, p. 388). Hence, sainthood can be recognized by its fruits.

> "Obedience to the truth purifies the soul (cf. 1Pet 1:22), and 'the wisdom from above is first pure, then peaceable, gentle, willing to yield, full of mercy and good fruits, without a trace of partiality or hypocrisy' (James 3:17)". (TT, art. 93)

In short, media celebrities are the products of an industry which satisfies the demand for authenticity and heroism by simulating authenticity and heroism. Media offer celebrities to their audience in order to attract and draw attention to themselves—in contrast to saints who are mostly persecuted and usually face repression (even from the Church itself), only to be finally accepted for their perfection in the fulfillment of the greatest commandment (Guardini [1981] 2007). Saints are also presented to believers as true manifestations of the face of Jesus so that believers can follow them and find their way to God. Hence, in this light, celebrities more closely resemble "anti-saints" than "saints".

### 4.9. Individualism

Individualism is a generally accepted way of thinking in modern societies. The theoretical concept of a fundamental subjective turn has taken root in society and culture but also in religion: a radical turn toward one's "self" which represents a new "sacrum" and which is nourished by mediated celebrity culture but also by religion itself (Ward 2019). In some theoretical concepts, individualism appears as the only bonding agent capable of uniting contemporary society. According to E. Durkheim ([1898] 1994, p. 70) the "religion of the individual" is, like all known religions, a social institution. He saw the religion of the individual he sees as the only value system that can unite a modern society and that therefore the practice of its cult is necessary to the social order. Drawing on these theories, some media scholars point to media as a "shrine of the cult of the individual" (Rothenbuhler 2005); they also point to an individualized "online religion" tailored to the needs of specific individuals (Hoover and Park 2005).

Theology, on the other hand, emphasizes the image of a triune community inviting man to love; encouraging him to transcend himself and sacrifice his self. "He who does not live for himself actually finds himself" (Küng [1974] 2007, p. 444). This is an extrapolation of the words of Jesus: "He who has found his life will lose it, and he who has lost his life for My sake will find it" (Matthew 10: 39). Whereas the concept of self-sacrifice as conceived within the media cult represents the highest degree of authenticity (Rothenbuhler 2005), in theology self-sacrifice as exemplified in Christ's sacrifice on the cross, represents the highest form of sainthood (Krapka 2000).

In summary, there is a direct conflict between the perception of individualism in media and religion. On that note, J. Ratzinger expresses his concern that "if man ( . . . ) is only an object to himself, ( . . . ) what will be man's attitude toward man when he can no longer find anything of the divine mystery in the other, but only his own know-how?" (Ratzinger [1968] 2007, p. 11). J. R. de Oliveira observes a dialectical relationship between the media and religion: it brings both positive and negative points in a growing duality. The first might make the religious individual more alive and engaged in his/her religious practices, but the latter leads to a consequent distancing of the religious individual from his/her religious practices and his/her faith (de Oliveira 2018, pp. 26–27).

## 5. Conclusions

The aim of this paper was to answer the initial question as to whether there are any essential and intrinsic similarities between media and religion, and also to examine whether there is such a thing as media religion. We have analyzed the subject from the perspective of Christian theology by focusing on individual areas which have already been subject to theoretical reflection.

At the outset of this study, we made a reference to the confrontation of the term "religion" and related terms (such as faith, God, rite, liturgy, unity, community, sainthood) which are used in media anthropology theories from the perspective of conservative Christian (Catholic) theology. We have provided the arguments and findings on the subject above. An audience rooted in Catholic theology can obtain a fuller understanding of how the theological terms have been used in media/celebrity studies including the identified shifts in comparison to their original (theological) meanings. On the other hand, an audience with a background in media (celebrity) studies can learn about the perspective of Catholic theology on the terms employed by media scholars when they reflect upon the subject in their studies, in addition to how the shift in their meanings is perceived from a Catholic perspective.

The findings show that media religion is, first, a religion without God, that is, a non-theistic religio, with the key capability of ensuring social cohesion. It has no real transcendent dimension and provides no place for faith in the sense of a human decision, that is, as a response to God's calling, subjection to God and an intense personal relationship with God. Several key considerations—the absence of verticality, a lack of transcendence toward eternity and the acceptance of the Revelation—have determined most of the partial conclusions presented herein.

Although media emotions generated through mass media ceremonies are powerful, they are only momentary and create ad hoc communities that remain essentially ephemeral. This is, of course, incomparable to the religious unity found in the reflection of the triune relationship through the gathering around the Father with Jesus Christ in the love of the Holy Spirit. Furthermore, the ephemeral dopamine effect of media shifts the focus from "experiencing" to "showing". Media liturgy satisfies the need for continuity and intimacy and distracts man with "small-scale transcendences" while failing to offer either a true object (God) or a transcendentally justified unity (brotherhood). Media are unable to attain the religious level of "ontologically most intense encounter" and appear to be limited to the form. Media cosmology attempts to explain the complexities of the world, but, unlike religion, it prefers the up to date, the faddish and the ephemeral to the eternal. Philosophically, it is underpinned by a relativism of values—something which contradicts the religious acceptance of the revealed Truth. It offers a "self-understanding" of the world and turns to man for the answers to fundamental questions about himself, refusing to accept the "top-down" conception of man and the world. Religious archetypes employed by the media do not flow from the media as such—they do not originate from a kind of "media-driven story of redemption"—and they are thus unable to compare to the history of God's presence among men. The archetypes which they employ are simply taken and borrowed from the religious sphere. Media celebrities are "anti-saints"; they have been elevated into the pantheon not on the basis of their perfect obedience to God's will or by a radical decision made out of love to "forfeit one's life", but rather due to their absolute adherence to an individualistic idolatry of the self.

As has been illustrated above, it is more appropriate to speak of the essential differences which exist between media and religion rather than of any similarities between the two. Media may be able to simulate the satisfaction of some of the religious needs of human beings, but the content which they offer is merely a form devoid of substance. Media lack the upward movement (towards God); instead, they lead man in a circle (back toward man). Indeed, one can speak of the cult of media—similar to the cult of man known from the pagan world—in which man becomes either the recipient (individualism) or the creator (media-centrism). On the other hand, "the journey of the heart" as suggested by M. Gavenda, offers authentic religious liberation from this entanglement, in that the irreversible change driven by new media technologies—which constantly shift our perception from the conceptual to the visual or emotional—is put to work in favor of human beings in order that they can turn their attention to God. Within such a concept, emotions become mechanisms that lead them to contemplation and allow them to come closer to their essence.

Media and celebrity theories quite naturally use the terms and narratives adopted from the sphere of religion. However, any effort to identify media (celebrity culture) with religion remains questionable. P. Ward makes a conciliatory suggestion that celebrity culture is too frivolous to represent religion as media fails in all of the basic functions of traditional religion; he also suggests that "real religion

resides with the worship of God and this religion is the only one that is to be trusted" (Ward 2017, p. 9). However, instead of refuting the religious characteristics of these processes, one should rather speak of something "akin to religion"—as is encapsulated in the term "parareligion". It is in this domain that the deified "self" is worshipped; "to speak of celebrity culture as a kind of theology, therefore, does not tell us anything about the Christian God, but it does reveal something of how we see ourselves". (Ward 2011, p. 6).

**Funding:** This research received no external funding.

**Conflicts of Interest:** The author declares no conflict of interest.

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
