# Peer review of "Media as Religion. Stardom as Religion. Really? Christian Theological Confrontation"

_religions, doi:10.3390/rel11110568_

Round 1
Reviewer 1 Report
There is interesting potential in an article focused on the religiosity or religious dimensions of media. And there are moments when this article succeeds along these lines. But overall, its strong normative and confessional theological orientation seems inappropriate to its topic. The author tries to account for this normative frame early on by stating that they are focusing their analysis along the lines of Christian theology rather than 'religion' in general. This is certainly a fair and appropriate move to make. But it's fair if operating in the distinction between a Christian and non-Christian theological paradigm, not Christian vs. the status of 'religion' in scholarly discourse, period. The author seems unfamiliar with the 'study of religion' types of questions that would comprise the heart of claims like the ones he/she/they want to make.
As a result, by the end of the piece the increasingly normative and confessional thrust comes to undermine the scholarly integrity. It remains unclear why the potential religiosity of media should, in the end, be evaluated by Christian standards and what the scholarly value of doing so might be.
It should be noted that the normative confessional orientation itself is very narrow, and presumes agreement on the part of Christian theologians around what the relationship between faith, belief, and religion even are (see esp. 238-240). It then evaluates the religiosity of media by the criteria of this narrow definition. But it remains unclear why the religiosity of media should be evaluated by this standard.
Finally, while the article also draws on a diversity of sources, it doesn't grapple with any of them in depth. As a result, it lacks a sustained argument and functions mostly at the level of a list of examples. Again, some of these examples are really interesting. But they need something to hold them all together.
I know that Pete Ward is a more confessional scholar working in the field of media and celebrity studies, so I'd recommend that this author check out Ward's work before proceeding.
For these reasons, I do not recommend this article for publication without a significant (and likely too substantial) rewrite.
Author Response
Dear reviewer,
thank you very much for your time and your helpful review.
At the same time, I ask you to kindly overlook my "rough" level of English. While I have my paper translated by a professional translator, this answer I am writing directly in English.
Thank you in particular for drawing my attention to the work of P. Ward. I specialize more in media studies and I reflect on celebrity studies only marginally, but his books and articles were very inspiring for me and I incorporated them into my paper in several places (especially his concept of parareligion and the concept of a fundamental subjective shift in culture, society, and religion + several other stimuli concerning the religious language in celebrity worship, the decline of the formal religion, etc.).
I responded to the argument about normativity, confessionalism, and the narrow Catholic focus of my approach directly at the beginning of the text, where I tried to explain more clearly why this particular angle of view is important and fruitful. My intention was just to compare with the major conservative Christian theological current, which I explain in more detail in the text. I consider it necessary and useful. Sure, a general religionist comparison would be interesting, but it would be a different work and its scope would be different (it is a topic for a book…).
I believe that (also thanks to P. Ward) I managed to sharpen and deepen my argument, also by adding a final paragraph in which I lean towards his conciliatory theory of parareligion. I understand that, overall, my text can make a "heterogeneous" impression, but it is my style of analysis and reasoning: I have tried to categorize the ideas and gradually get closer, so the text may have too many subtitles. However, I believe that with careful reading, the structure and system will emerge. To this end, I have added some ideas to better highlight the mainline of Christian evaluation of the terms that media and celebrity studies borrow from religious language.
Thank you again for your assessment effort and help in refining my text.

Reviewer 2 Report
This paper is an effort to work with two very different vocabularies--that of media studies and media anthropology and what we would ordinarily think of as "religion" ( in this case very specifically Western Christian religion
These are very different vocabularies and perspectives and the effort does not always work. Or, to be more precise, it is not clear what it would mean for it to work: It is not clear (at least not to me)When and how and in what ways would the vocabulary of one help in the analysis of the other?
The author's goals seem well beyond the empirical: for example, how does the reliance of religious groups on web sites and mass media affect the content and practice of their beliefs, the ways in which community might be created or sustained?
We don't get that. what we do get is a sometimes tantalizing confrontation of metaphors, of qualities of experience or of belonging.
the author is also a little loose with terms like "man" --see the last two sentences
also with references to the "function" of this or that. Line 412 on the function of the media.. I did not know it had a definable function--an effect perhaps, but a function?
i would like to be more convinced by this than i am
the idea is tantalizing but for me incomplete
i think you need also a review from someone who knows the area of media anthropology, which I do not
Author Response
Dear reviewer,
thank you very much for your time and your helpful review.
At the same time, I ask you to kindly overlook my "rough" level of English. While I have my paper translated by a professional translator, this answer I am writing directly in English.
Thank you in particular for your suggestion regarding two different "vocabularies". I developed my position and explained it better in footnote no. 1. At the same time (especially in the introduction itself) I explained in more detail the need and my intention to look at this problem precisely through the lens of Christian theology (in the sense of the major official conservative current).
As for "empirical goals", my work has no empirical character or any ambitions of this kind, on the contrary, it is a theoretical study in which I refer to previous empirical and theoretical research and findings, summarize them, systematize, create categories, structure - to I could build my comparison on that.
Thank you also for your attention to terminological matters. The term "function" originated from the shift of E. Rothenbuhler's original statement, I changed it to "activity", which corresponds to Rothenbuhler's original. The term “man” I replaced, for the sake of clarity, with the term "human being", which, inter alia, has no gender and is perceived less archaically. But in the context of traditional religious literature, I perceive the term "man" as quite common, see e.g. John Milton in Paradise Lost, who wants to "justify the ways of God to men" ...
Thank you again for your assessment effort and help in refining my text.
Reviewer 3 Report
The paper is generally well-written but could be sharpened. It actually is a critique of what it says is an increasing tendency within celebrity studies to identify the media with religion.
Although I am in a communication department and do work on religion, I am not so familiar with the literature on celebrity studies. My problem with the paper is that I find the position they are arguing against to be palpably ridiculous.
One thing the authors need to do is show better that this identification is actually being made. Does Derrida really think television is a form of religion? We need more than just a citation here. There is a difference too between saying that various aspects of profane existence draw on religious elements and saying that they actually are religious. We have rites and rituals throughout everyday life: greetings, acknowledging status changes and so forth. Blessing people who sneeze. We have cultural heroes. So is what is being attacked saying no more? If so, there is not much of a target.
I don't see anythng particularly liturgical about watching television, and nor do I see the category of emotions as a point of similarity.
If Couldry speaks of the media as offering a world center, that does bear some similarity to what Mircea Eliade describes as a religious axis mundi. But how far is Couldry taking this?
My point is that the target of attack here appears too weak to merit an attack. The authors do a lot of stage setting that could be eliminated in place of presenting a more coherent foil of attack. We need to see a fuller argument made by these people to be able to tell what they are up to. There is too much jumping around.
I also find too sketchy the authors' treatment of religion. They take us through a number of different thinkers but then seem to settle too quickly on a divinity, despite their observation that some religions have none. Nor do I think that "faith" is unpacked so well. What does Tillich mean by that term? Certainly not what an Evangelical Christian would.
Author Response
Dear reviewer,
thank you very much for your time and your helpful review.
At the same time, I ask you to kindly overlook my "rough" level of English. While I have my paper translated by a professional translator, this answer I am writing directly in English.
Thank you in particular for pointing out the need to sharpen my argument. To this end, I have added a new paragraph at the beginning and at the end, where I am trying to explain my position more clearly. In this, I was also helped by the ideas of P. Ward, whose concepts related to my topic I incorporated in several parts of the text. Especially useful for me was his concept of parareligion and fundamental subjective turn in society, in the media, and in religion (+ inspiring ideas concerning the decline of formal religion, the use of religious language in celebrity worship, etc.). I believe that my concept is now clearer.
I understand that, overall, my text can make a "heterogeneous" impression, but it is my style of analysis and reasoning: I have tried to categorize the ideas and gradually get closer, so the text may have too many subtitles. However, I believe that with careful reading, the structure and system will emerge. To this end, I have added some ideas to better highlight the mainline of Christian evaluation of the terms that media and celebrity studies borrow from religious language.
I work by the method of theoretical study, not empirical research, so I draw all forms of penetration of religion and media from established theorists, summarize them, categorize them, so that I can then subject them to comparison and evaluation. The liturgy or emotions mentioned by you may, of course, appear to anyone subjectively to be meaningless in this context, but I refer, analyzing them, to previous works in the field.
I returned to my references to Derrida and Couldry. As far as Derrida is concerned, I am not saying that he "thinks that television is a form of religion", there is not such a sentence in the text, I mentioned Derrida only once, in the introduction, where I stated that we can "ask with him if there are intrinsic similarities between religion and the media, whether there is a media religion”. Because it was a question that interested Derrida, but he didn't say it was. In this version of the text, I supplemented his quote from 2001, where he specifically comments on the internal connection between the media and Christianity (unlike other Abrahamic and other religions).
Similarly, I extended my appeal to N. Couldry. He does not respond directly to Eliade, but to clarify his argument concerning the "social center" mediated by the media, I have inserted a more detailed explanation in the text with a more extensive direct quotation.
As for the narrow orientation of my media-theological comparison to theistic religion, specifically to Christianity, I explained this position in the introduction. At the same time, I made it clear that I do not understand my text as a controversy, not as an attack, but as a curious question, inquiry, looking at one through the lens of another. I consider this approach to be very fruitful (as I also explain in footnote no. 1). My intention was just to compare with the major conservative Christian theological current, which I explain in more detail in the text. I consider it necessary and useful. Sure, a general religionist comparison would be interesting, but it would be a different work and its scope would be different (it is a topic for a book…).
Thank you again for your assessment effort and help in refining my text.
Round 2
Reviewer 1 Report
The article is improved, and I appreciate the author's work in this regard. I also agree with their point that it is appropriate to conduct their analysis according to a particular Christian theological tradition (in this case, Catholicism and, arguably, conservative Catholicism). But there remain two issues with how this specificity is carried out in my view.
First, the author continues to use 'theology,' 'Christianity,' and 'religion' as synonyms for 'Catholicism'. But these are not all the same thing. The argument needs to be tightened so that it only makes claims out of Catholic theology for Catholic theology. Claims made with Catholic theology cannot be automatically generalized to Christianity, theology, or religion in general.
My second point is connected, but geared to the question of audience. If the author is to use Catholic theology in a normative way (which they do), it would be more appropriate to narrow the article's conclusions to having important for only a Catholic theological audience. The normative thrust moves the article out of being directed towards a media studies audience. I think it's fine to gear it towards specifically Catholic theological conclusions, but it needs to be edited towards that end.
Were the article to fit within a media studies orientation, then it would be more appropriate to use Catholic theology to reveal how media tropes certain forms of 'religion,' how it depends on them and redeploys them in ways that both connect with and completely re-arrange and re-imagine what it is they trope. The article could be edited towards these ends. But it seems that this is not the argument that the author wants to make. Fair enough! But I don't think that the argument they are making is a 'media studies' argument.
This is likely a question of 'fit.' If this journal is open to publishing a more normative Catholic theological article, then the article works if it does light edits to tighten up the argument as I note above. If it isn't, then the article requires a much larger scale further editing.
Some stray concerns:
1. the section on ritual and liturgy presumes a traditional television model. Its conclusions to do not work for streaming services, DVR'd programming, and the myriad ways in which we actually access tv right now.
2. Cosmology -- in the first half, this is where the normative thrust comes through most forcefully.
3. Cult of individualism -- if anything this reveals media's fit with the turn to the subject in mainstream Christian practice
4. Does this journal require inclusive language? If so, then all references to 'man' and 'mankind' need to be changed to language of 'humanity'
5. There is slippage between tv and media throughout -- it would be helpful to get that language consistent to avoid over-abstract claims.
Author Response
Dear reviewer,
Thank you very much for your precise and helpful comments!
To the terms theology, Christianity, religion, and in connection with the need for claims made with Catholic theology not to be automatically generalized to Christianity, theology, or religion in general, I have added a paragraph in the introduction (first green paragraph).
Concerning the audience and the need to specify to whom the conclusions of the study are addressed, I have added a paragraph at the end (third green paragraph).
I responded to the remark about the slippage between tv and media throughout in a separate paragraph in the section on the structural similarities between the media and religion, following Thomas' quote (second green paragraph).
Regarding the inclusive language, I will follow the editorial instructions.
All the text has undergone native proofreading, so I hope that it would be more “fluent” now.
Thank you very much once more!
Reviewer 3 Report
This piece is much improved. The additions do a lot to present the tendency being attacked as a much more worthy target. I must say I was not optimistic that the author would be able to turn the paper around so well. So I especially commend the work done here.
I see the author is not a native English speaker and has had the paper edited by someone who is. I think the writing is also much improved thereby. I still think it needs some going over for run-on sentences and the like. But these are now minor concerns.
Author Response
Dear reviewer,
Thank you very much for your comment and your commendation for my improving work.
All the text has undergone native proofreading, so I hope that it would be more “fluent” now.
Thank you very much once more!
